# Promising Natural Remedies for Alzheimer’s Disease Therapy

**DOI:** 10.3390/molecules30040922

**Published:** 2025-02-17

**Authors:** Amin Mahmood Thawabteh, Aseel Wasel Ghanem, Sara AbuMadi, Dania Thaher, Weam Jaghama, Donia Karaman, Rafik Karaman

**Affiliations:** 1Department of Chemistry, Birzeit University, West Bank, Ramallah 00972, Palestine; athawabtah@birzeit.edu; 2Faculty of Pharmacy, Nursing and Health Professions, Birzeit University, West Bank, Ramallah 00972, Palestine; aseel.wasel@gmail.com (A.W.G.); abumadisarah@gmail.com (S.A.); daniathaher2000@gmail.com (D.T.); weam_rezeq@yahoo.com (W.J.); 3Pharmaceutical Sciences Department, Faculty of Pharmacy, Al-Quds University, Jerusalem 20002, Palestine; kdonia65@yahoo.com; 4Department of Sciences, University of Basilicata, Via dell’Ateneo Lucano 10, 85100 Potenza, Italy

**Keywords:** Alzheimer’s disease, anti-amyloid drugs, phytoconstituents, *Ginkgo biloba*, rosemary

## Abstract

This study examines the intricacies of Alzheimer’s disease (AD), its origins, and the potential advantages of various herbal extracts and natural compounds for enhancing memory and cognitive performance. Future studies into AD treatments are encouraged by the review’s demonstration of the effectiveness of phytoconstituents that were extracted from a number of plants. In addition to having many beneficial effects, such as improved cholinergic and cognitive function, herbal medicines are also much less harmful, more readily available, and easier to use than other treatments. They also pass without difficulty through the blood–brain barrier (BBB). This study focused on natural substances and their effects on AD by using academic databases to identify peer-reviewed studies published between 2015 and 2024. According to the literature review, 66 phytoconstituents that were isolated from 21 distinct plants have shown efficacy, which could be encouraging for future research on AD therapies. Since most clinical trials produce contradictory results, the study suggests that larger-scale studies with longer treatment durations are necessary to validate or refute the therapeutic efficacy of herbal AD treatments.

## 1. Introduction

Memory loss, impaired executive function, and slow cognitive decline are the hallmarks of AD, the most common type of dementia and a degenerative neurological disorder. The frequency of AD is expected to increase dramatically as life expectancy rises worldwide, and it typically affects older people. The pathological hallmarks of AD are the buildup of tau tangles and amyloid-beta plaques, which result in neuronal death and dysfunction. Patients complain of cognitive decline, which is made worse by oxidative stress, chronic inflammation, and synapse loss [1,2]. In addition to its cognitive effects, AD has a major negative influence on patients’ emotional and physical health, which puts a heavy strain on caregivers and healthcare systems around the globe. Over 55 million people worldwide are estimated to have dementia, with AD responsible for 60–70% of these cases [3]. There is an urgent need for new therapeutic approaches as, despite much study, the effectiveness of current pharmaceutical treatments is still limited and focuses mostly on symptoms rather than the progression of the disease [4]. The growing prevalence of AD emphasizes the significance of investigating novel approaches to treatment and prevention, especially with regard to natural products and alternative medicines [5].

With a focus on their mechanisms of action, preclinical and clinical evidence, and potential as therapeutic alternatives or adjuncts to current medications, this study aims to provide a thorough overview of the most promising natural products for the treatment of AD. Natural substances that may target several facets of AD pathology, such as amyloid aggregation, tau hyperphosphorylation, oxidative stress, and neuroinflammation, are gaining attention due to the drawbacks of conventional drug therapies like acetylcholinesterase inhibitors and amyloid-targeting monoclonal antibodies. Along with addressing the issues of dosage and bioavailability, this study also seeks to highlight important natural items that have demonstrated potential in modifying certain disease pathways, such as curcumin, *Ginkgo biloba*, and *Ocimum tenuiflorum*. Furthermore, the study aims to assess the present status of research and clinical trials and investigate the possibility of incorporating these natural compounds into current AD treatment plans. Generally speaking, the objective is to present a fair assessment of the medicinal potential of natural items, promote additional research, and highlight prospects for their incorporation into upcoming AD treatment plans.

### 1.1. Current Treatment Strategies and Their Limitations

The goal of current AD treatment options is to control symptoms rather than delay or reverse the progression of the illness. The most typically given drugs are acetylcholinesterase inhibitors, which include galantamine, rivastigmine, and donepezil. They function by raising the brain’s acetylcholine levels, improving neuronal transmission, and momentarily lessening cognitive symptoms [6]. Memantine and other members of the N-methyl-D-aspartate (NMDA) antagonist drug family are used to treat memory and cognitive issues by modifying glutamate activity [7]. Monoclonal antibodies that target and try to remove amyloid-beta plaques, a defining feature of AD pathology, such as lecanemab and aducanumab, have just received FDA approval. The therapeutic advantages of these medications are still debatable, though, as some studies have found little impact on the course of the disease [6,8]. Notwithstanding their benefits, these treatment alternatives only slightly improve symptom management and do not address the underlying, as yet unidentified, causes of AD. Additionally, these medications may cause gastrointestinal problems or, in the case of amyloid-targeting treatments, hemorrhage and edema in the brain [9]. Therefore, there is a growing need for stronger treatments that can stop or reverse the progression of the disease; ideally, these treatments should target the complex pathophysiology of AD, which includes tau protein aggregation, oxidative stress, and neuroinflammation [9,10].

### 1.2. Rationale for Exploring Natural Products as Potential Treatments

Investigating natural products as potential treatments is justified by their wide range of chemical composition, long history of usage in traditional medicine, and capacity to target many pathogenic processes associated with AD. The complex mechanisms of action that natural products often exhibit, such as antioxidant, anti-inflammatory, and neuroprotective qualities, are well matched with the multifaceted nature of AD pathogenesis, which includes amyloid plaque accumulation, tau tangles, oxidative stress, and neuroinflammation [11,12,13]. Numerous natural substances are thought to change important biochemical pathways involved in neural defense and repair, providing a viable substitute for the symptomatic therapies now in use [12,13]. Furthermore, compared to traditional pharmaceuticals, which might result in negative reactions including gastrointestinal distress or neuropsychiatric symptoms, natural products are frequently seen to be safer and have fewer side effects [14]. As a result, research on AD has become more interested in finding plant-based or other natural compounds, such as polyphenols, flavonoids, and alkaloids [15]. Additionally, new research indicates that when used in conjunction with conventional medicines, natural products may provide supplemental or adjunctive advantages that could increase efficacy and improve patient outcomes [15,16]. Notwithstanding the encouraging potential, additional clinical verification and investigation into bioavailability, dosage, and long-term safety are essential to converting these natural remedies into effective AD therapy alternatives [17].

The selection criteria for natural treatments for AD include mechanisms of action, wherein treatments must focus on important clinical characteristics of AD, including oxidative stress, tau hyperphosphorylation, amyloid-beta plaque formation, and neuroinflammation [18,19]. Additionally, they must demonstrate particular biological properties such as neuroprotective, anti-inflammatory, and antioxidant actions [19]. On the other hand, several treatments are selected because of their lengthy history of usage for neurological and cognitive benefits in traditional medical systems (such as Chinese medicine and Ayurveda) [3,5].

Another criterion for choosing natural treatments for AD is scientific proof. Remedies are thus chosen based on preclinical and clinical research showing effectiveness in modifying AD pathways [13,14]. The sources here included peer-reviewed research from 2015 to 2024. Additionally, a natural treatment should have a positive safety and side-effect profile when compared to conventional treatments, with a focus on older and less dangerous cures [7,8]. Furthermore, natural compounds with bioavailability in the brain and the ability to effectively cross the blood–brain barrier are valued for their potential to complement or operate in concert with current AD treatments [20,21].

## 2. Mechanisms of AD

### 2.1. Pathophysiological Features of AD

Tau tangles, amyloid plaques, neuroinflammation, and oxidative stress are among the pathophysiological features of AD; the development of cognitive decline is accelerated by a complicated and self-reinforcing loop that includes mitochondrial dysfunction, synapse loss, and neuronal death [18,19,20]. Each of these processes is crucial to the onset and progression of the disease, and when combined, they create a toxic stress environment that results in irreversible neuronal death. Recent years have seen a significant increase in our understanding of these pathways, and there is growing evidence that the disease is influenced by their intricate connections [2,15,21,22,23]. In Figure 1, the pathophysiology of AD is described.

#### 2.1.1. Amyloid Plaques and Tau Tangles

The buildup of amyloid-beta plaques in the brain is one of the most well-known pathological characteristics of AD. Under aberrant circumstances, amyloid-beta, a portion of the amyloid precursor protein (APP), builds up, and clumps form insoluble plaques. These plaques mostly develop in the brain’s extracellular spaces, especially in areas that are essential for memory and cognition, like the cerebral cortex and hippocampus [2,18].

By disrupting synaptic function and causing neuroinflammation, amyloid-beta plaques are believed to impair neuronal communication [19,20]. The activation of microglia, the brain’s resident immune cells, has also been connected to amyloid-beta plaques. These cells release pro-inflammatory cytokines that worsen neuronal damage when they are activated [2,21].

The pathophysiology of AD is characterized by tau tangles in addition to amyloid plaques. A protein called tau typically helps neurons’ microtubules to remain stable, but in AD, tau becomes hyperphosphorylated and starts to twist into tangled knots inside neurons. These tangles hinder intracellular transport, cause synaptic dysfunction, and ultimately result in neuronal death by interfering with microtubules’ normal activity [21,22]. The presence of both amyloid plaques and tau tangles is hypothesized to create a toxic environment that accelerates the degradation of neurons and the decrease in cognitive function [22].

#### 2.1.2. Neuroinflammation, Oxidative Stress, and Mitochondrial Dysfunction

A key factor in the pathophysiology of AD is neuroinflammation. Microglia and astrocytes are the main mediators of the inflammatory response that is brought on by the buildup of tau tangles and amyloid-beta plaques in the brain. By clearing away cellular debris and protecting the brain from infections, microglia normally aid in maintaining brain homeostasis. Reactive oxygen species (ROS), inflammatory cytokines, and other toxic chemicals are released by chronically activated microglia in AD, further damaging neurons [21,22,23,24]. Increased oxidative stress, the BBB’s disintegration, and the progressive loss of neurons are all caused by this neuroinflammatory reaction. Furthermore, the neuroinflammatory cycle may be exacerbated by the production of pro-inflammatory chemicals as a result of astrocyte activation, which aims to heal neuronal damage [25].

In the early stages of AD, neuroinflammation serves as a protective response; but, as time passes, it turns maladaptive and accelerates the neurodegenerative process. The buildup of tau tangles and amyloid-beta plaques is accelerated by the persistent inflammatory milieu, resulting in a vicious loop that speeds up the deterioration of cognitive function [26]. As a result, neuroinflammation is both a result of AD and a major factor in its development.

An additional important element in the pathophysiology of AD is oxidative stress, which is defined by the overproduction of ROS. Due to their high metabolic activity and weak antioxidant defenses, neurons are especially susceptible to oxidative injury. Neuronal malfunction and mortality result from ROS’s damage to lipids, proteins, and DNA, among other cellular components [21]. The buildup of amyloid-beta plaques in AD causes oxidative stress, which can lead to the production of ROS and further harm neurons [27]. Furthermore, tau hyperphosphorylation is encouraged by oxidative stress, which exacerbates tau tangle formation and compromises neuronal function [28].

One of the main characteristics of AD is thought to be mitochondrial dysfunction, which is directly related to oxidative stress. Cellular energy production depends on mitochondria, and when they malfunction, there is an energy deficit that affects synaptic plasticity and neural communication. Increased oxidative damage to the mitochondria in AD causes a reduction in ATP synthesis and the buildup of damaged mitochondrial components [29]. Because mitochondrial dysfunction damages cellular structures, increases oxidative stress, and impairs neurons’ capacity for self-healing, it is believed to have a role in the pathophysiology of AD. Because mitochondria are essential for preserving the health of neurons, their malfunction speeds up neurodegeneration and cognitive impairment in AD [29,30].

#### 2.1.3. Synaptic Loss and Neuronal Death

Neuronal death and synaptic loss are the final effects of AD’s pathogenic characteristics. The places of connection between neurons, known as synapses, are especially susceptible to tau disease and amyloid-beta toxicity. It has been demonstrated that amyloid-beta plaques affect synaptic plasticity, which is crucial for learning and memory and causes synapses to either strengthen or weaken in response to activity [20,21]. Furthermore, tau tangles cause microtubules to become unstable, which interferes with the delivery of vital proteins and nutrients to synapses and further impairs their functionality [20,21]. Neurons start to deteriorate as synaptic loss increases, especially in regions like the cortex and hippocampus that are essential for memory formation and cognitive function [1]. The atrophy of brain areas and the cognitive decline that characterizes AD are caused by the death of neurons. The clinical signs of AD, such as memory loss, disorientation, and poor executive function, are ultimately caused by this neuronal death [1].

### 2.2. Potential Targets: Inflammation, Oxidative Stress, Neuroprotection, and Amyloid Plaque Inhibition

Because of the complexity of AD’s pathophysiology, treatment approaches have not yet produced conclusive answers. However, a number of possible therapeutic targets have surfaced in recent years, with an emphasis on amyloid plaque inhibition, oxidative stress, inflammation, and neuroprotection. These pathways not only provide light on the mechanisms underlying the disease, but they also present encouraging opportunities for the creation of drugs that could delay, stop, or even reverse the course of the illness [2,5,8].

#### 2.2.1. Inflammation as a Therapeutic Target

In the context of AD, inflammation is becoming more widely acknowledged as a major factor in the development of the disease. A key factor in this process is the long-term activation of the brain’s resident immune cells, known as microglia. The overactivation of microglia in AD causes the release of pro-inflammatory cytokines and chemokines, which can worsen neuronal damage, even though their usual purpose is to defend the brain against infections and remove cellular debris [31,32]. Cognitive decline is exacerbated by this neuroinflammatory response, which damages neurons and interferes with synaptic function. Thus, focusing on neuroinflammation has become a viable treatment approach. Non-steroidal anti-inflammatory medications (NSAIDs) and other modulators of inflammatory pathways and inhibitors of microglial activation have been suggested as viable options. Nevertheless, conflicting findings from clinical trials examining NSAIDs’ impact on AD [33,34] have prompted additional research into more focused immune response modulators. One possible method for managing the neuroinflammatory aspect of AD is the use of biologics that target certain cytokines or immune receptors implicated in the inflammatory cascade, such as interleukin-1 (IL-1) [33,35].

#### 2.2.2. Oxidative Stress and Its Role in AD

As previously stated, oxidative stress contributes to the hastening of neurodegeneration. Oxidative damage, namely to lipids, proteins, and DNA, is caused by an imbalance between reactive oxygen species (ROS) and the brain’s antioxidant defenses in AD. It is thought that this damage worsens synapse loss, hinders neuronal function, and encourages the development of amyloid plaques [21,27].

Antioxidants like vitamin E and curcumin, as well as medications that can improve mitochondrial function and lower ROS generation, are examples of therapeutic strategies used to combat oxidative stress [28].

#### 2.2.3. Neuroprotection: Enhancing Neuronal Resilience

Neuroprotection is another promising strategy in the development of Alzheimer’s drugs, in addition to addressing oxidative stress and inflammation. The ability of neurons to tolerate external stimuli, heal damage, and continue to function is known as neural resilience. Restoring cellular homeostasis and preventing neuronal death are the main goals of strategies to improve neuroprotection [29]. Numerous important pathways that support neuroprotection have been found by research, such as the modulation of neurotrophic factors like brain-derived neurotrophic factor (BDNF), the activation of cellular repair processes, and the improvement of synaptic plasticity. Preclinical models of AD, for example, are investigating the use of substances that activate the Nrf2 (nuclear factor erythroid 2-related factor 2) pathway, a crucial regulator of antioxidant response and cellular protection [30]. Enhancing autophagy, a mechanism by which cells break down and recycle damaged proteins and organelles to stop the buildup of harmful chemicals in the brain, is another possible neuroprotective strategy [30]. Neuroprotective drugs are a new way to delay the progression of AD, even if they are still in the early phases of development.

#### 2.2.4. Amyloid Plaque Inhibition: Targeting Beta-Amyloid Aggregation

The buildup of amyloid plaques, which are mainly made of beta-amyloid peptides (Aβ), is one of the most noticeable symptoms of AD. These plaques are thought to be crucial to the pathophysiology of the disease because they impair neuronal function, cause neuroinflammation, and more. As a result, for many years, medication development has focused on amyloid plaque inhibition [2,18]. Several approaches, such as the use of enzyme inhibitors, small compounds, and monoclonal antibodies, have been investigated to prevent the formation of amyloid plaque. To target and remove amyloid plaques from the brain, monoclonal antibodies like aducanumab and lecanemab have been created. Despite debate regarding its clinical efficacy and high cost, the U.S. FDA authorized aducanumab in particular for the treatment of AD in 2021 [36]. Because they specifically address one of the disease’s defining characteristics, these treatments mark a substantial advancement in the treatment of AD. Nonetheless, the amyloid hypothesis is still debatable; some scientists doubt that removing amyloid plaque on its own is enough to stop or reverse the progression of the disease [37,38]. Recent studies have moved toward combination treatments that target several AD pathways, such as oxidative stress, inflammation, and amyloid plaques, in response to these worries. These methods seek to address the disease’s complex nature and offer more extensive treatment choices [38,39].

## 3. Phytochemicals for AD Treatment

Natural products have a variety of effects on neurodegenerative diseases like AD, including anti-inflammatory, antioxidant, neuroprotective, and neuroregenerative properties. These properties may target important elements of AD pathology, including tau hyperphosphorylation, oxidative stress, amyloid-beta accumulation, and neuroinflammation [40,41]. Furthermore, because of their generally moderate side-effect profiles, natural products are frequently thought of as safer substitutes or supplements to synthetic medications [41]. Finding particular bioactive substances from natural sources that can support or improve existing treatment approaches is becoming more and more popular as our understanding of AD processes advances [40,41,42]. The pressing need for disease-modifying therapies that not only reduce symptoms but also halt or reverse the course of the illness is driving this shift toward investigating natural compounds for AD therapy [42]. Exploration of these chemicals offers hope for the development of new, affordable, and effective therapies for AD because of the enormous variety of possible natural compounds [43].

### 3.1. Ashwagandha

Ashwagandha (*Withania somnifera*), a plant noted for its medicinal properties, contains around 35 phytochemicals, including steroidal lactones and alkaloids. Withanone is the predominant steroidal lactone, and withanine is the primary alkaloid. Withaferin A, withanolides A–Y, withasomniferin A, withasomidienone, and withasomniferols A–C are further alkaloids. Ashwagandha may have anti-Alzheimer’s properties, according to studies on its phytoconstituents [44,45,46]. Major alkaloids like withanone have been shown in recent studies to increase pro-inflammatory cytokine levels by inhibiting amyloid β-42 and increasing the activity of enzymes including glutathione, acetylcholine, and secretase. Furthermore, because of its high binding affinity, withanolide-A inhibits human acetyl cholinesterase. According to experimental studies, APP/PS1 transgenic mice and APPSwlnd mice treated with semi-purified ashwagandha root extract (which mostly contains withanolides and withanosides) exhibited neuroprotective benefits against H2O2- and β-amyloid cytotoxicity, hence reversing AD [44,47,48].

Additionally, ashwagandha protects rats from propoxur exposure. While the aqueous extract of ashwagandha containing derivatives of withanolide, like withaferin A, shows potential protection in differentiated pheochromocytoma PC12 cells infected by hydrogen peroxide and Aβ cytotoxicity, the methanolic chloroform extract of ashwagandha elicits high cell viability and enhances PPAR-c levels [49,50]. The primary phytoconstituents of ashwagandha and their corresponding therapeutic effects for AD are summarized in Table 1 and Figure 2.

### 3.2. Rosemary

Rosemary (*Rosmarinus officinalis*), a Mediterranean-originated shrub with a wide range of uses, belongs to the Lamiaceae mint family. Two phenolic diterpenes found in rosemary, rosmarinic acid and carnosic acid, have attracted much attention among the plant’s many phytocompound components due to their distinct biological characteristics [57]. It has been shown that ceramides, such as cysteine 151, which are directly S-alkylated on Kelch-like ECH-associated protein 1 (Keap1), shield brain cells against harm caused by cyanide. They have also been shown to protect Cellosaurus cell line PC12h cells from reactive-oxygen-species-mediated damage by selectively expressing phase 2 antioxidant enzymes, such as heme oxygenase-1 (HO-1), NADPH quinone oxidoreductase 1 (NQO1), and c-glutamyl cysteine ligase (c-GCL), which alter the redox status inside the cell [58].

In mice given rosmarinic acid (found in *Melissa officinalis*, *Perilla frutescens*, and *Salvia officinalis*), the cerebral cortex exhibited a reduction of Aβ accumulation and elevated levels of monoamines, including dopamine, levodopa, norepinephrine, and 3,4-dihydroxyphenylacetic acid. It also demonstrated the dopamine-signaling pathway by significantly inhibiting MAO-B in the substantia nigra and ventral tegmental area [59]. It directly inhibited Aβ40 and Aβ42 proliferation, elongation, and instability [60]. In this study, 23 human patients (12:11, treatment group: placebo group) with mild AD-related dementia were treated with an extract of *Melissa officinalis* that included rosmarinic acid. The study was double-blind, randomized, and placebo-controlled. The Neuropsychiatric Inventory Questionnaire score (NPI-Q) improved by 0.5 points, indicating a significant difference (*p* = 0.012) [51,60].

Nepitrin flavonoid, produced from *R. officinalis*, inhibited acetylcholinesterase activity (IC50 value: 65 μg/mL) and BChE (IC50 value: 72 μg/mL) in a concentration-dependent manner while improving memory in Swiss albino mice and reversed the amnesia induced by scopolamine. Nepetin binds to the same binding site (Trp279, Phe330, and Tyr334) as donepezil, a typical medication for AD, and creates a comparable set of interactions with it, along with a few more (Arg289 and possibly Trp84), according to molecular docking studies [61,62]. It has been demonstrated that neophytadiene and a number of compounds present in *Rosmarinus officinalis*, including 1,8-cineole and β-caryophyllene (listed in Table 2 and Figure 2), are acetylcholinesterase inhibitors and protect PC12 cells from oxidative damage.

**Table 2 molecules-30-00922-t002:** Rosemary’s phytoconstituents and their respective properties for treating AD.

Bioactive Compound	Structure No.	Investigation Method	Mechanism of Action
Rosmarinic acid [57]	**8**, in Figure 2	In vivo (Tg2576 mice)	Reduces Aβ buildup while increasing levels of dopamine, levodopa, norepinephrine, and 3,4-dihydroxyphenylacetic acid in the cerebral cortex. Down-regulated MAO-B in the substantia nigra and ventral tegmental region.
		In vitro	Prevented the development, growth, and destabilization of Aβ40 and Aβ42.
Carnosic acid [58,59]	**9**, in Figure 2	In vitro Embryonic Sprague-Dawley rats’ cerebrocortical neurons and astrocytes.	Activate the Keap1/Nrf2 pathway by directly S-alkylating certain cysteines (such as cysteine 151) on Keap1 to protect brain cells against cyanide-induced damage.
		Primary neuronal/glial mesencephalic cultures derived from C57BL/6J mouse fetuses.	
		Human iPSCs were generated from fibroblasts. In vivo by NSA mice	
Nepitrin [51,60]	**10**, in Figure 2	In vivo (Swiss albino mice)	Swiss albino mice given scopolamine exhibited improved memory and concentration-dependent inhibition of BChE (IC50: 72 μg/mL) and acetylcholinesterase (IC50: 65 μg/mL).
		In silico	With a few special modifications (Arg289 and perhaps Trp84), it causes a similar set of interactions to donepezil, a common AD drug, and occupies the same binding location (Trp279, Phe330, and Tyr334).
Neophytadiene [51,60]	**11**, in Figure 2	In vitro	Tyrosinase, acetylcholinesterase, and BChE are all inhibited.
		In silico	Inhibits tyrosinase.
Eucalyptol [51,60,61]	**12**, in Figure 2	In vitro (cells (SH-SY5Y))	Prevents Aβ42 oligomerization and reduces iron-dependent ROS generation.
		In vitro (cells (PC12))	Reduces mitochondrial membrane potential, inhibits acetylcholinesterase, stops Aβ42 oligomerization, stops oxidative damage, and blocks NF-κB activation caused by Aβ25–35, NOS-2, and COX-2.
β-caryophyllene [51,62]	**13**, in Figure 2	In vivo (APP/PS1 AD mice)	Decreases astrogliosis, microglial activation, levels of COX-2, and the mRNA levels of the pro-inflammatory cytokines TNF-α and IL-1β in the cerebral cortex, due to the activation of cannabinoid receptor 2 and the PPARγ pathway, which also lowers the neuroinflammatory response.

### 3.3. Ginkgo biloba

Although ginkgo leaf extract has long been used to treat memory and blood problems, it is currently the most researched herbal remedy for AD. Increased ginkgo consumption is known to improve mood, energy, and memory [63,64]. Supplementing with ginkgo showed the least decrease in MMSE scores among people over 65 who were not demented. Thus, after 20 years of research, a significant discovery was made, and the positive effects of ginkgo on cognitive function are now widely recognized [63,64,65].

One investigation involved injecting β-amyloid aggregates, ginkgo extract (EGb-761), and hyperbaric oxygen into an animal model of AD [66]. The EGb-761 extract contained 2.8–3.4% ginkgolides A, B, and C, 6% terpene lactones, including 2.6–3.2% bilobalide, 0.8% ginkgolide B, and 3% bilobalide, and 25% flavone glycosides, including isorhamnetin, kaempferol, and quercetin [67,68]. When EGb-761 and hyperbaric oxygen were combined, the Morris water maze test showed notable improvements in memory and cognitive function. Additionally, the nuclear factor kappa-B pathway was activated, which decreased hippocampus cell death. Hyperbaric oxygen therapy in conjunction with ginkgo extract has been shown to be more effective than monotherapy [67,68,69].

A second investigation employing APP-transgenic TgCRND8 mice assessed the effects of administering gingko EGb-671 extract over a prolonged length of time [69,70]. By preventing β-amyloid activation of microglia, gingko extract has been shown to minimize cognitive impairment, decrease synaptic damage, activate autophagy, and regulate neuronal inflammation. By lowering mitochondrial dysfunction, gingko leaf extract has been clinically shown to improve cognitive function in older persons without dementia, vascular dementia patients, and AD patients. This validates the mitochondrial cascade theory of dementia [70]. EGb-761 has been shown in numerous trials to be effective in improving functioning, behavior, and cognition in individuals with mild cognitive impairment who also have neuropsychiatric symptoms [67,68,69,70]. EGb761 has shown promising results in numerous trials with practical applications. The Geriatric Evaluation by Relative’s Rating Instrument (GERRI) and the AD Assessment Scale—Cognitive subscale (ADAS-Cog) showed improved performance among treatment groups with AD or multi-infarct dementia. The GERRI score was enhanced by 0.14 points (*p* = 0.004), and the ADAS-Cog score improved by 1.4 points (*p* = 0.04), when compared to the placebo group; for example, at dose = 40 mg/d, N = 327, the treatment group improved by 0.05 points (*p* = 0.09), while the placebo group’s GERRI score decreased by 0.07 points from its baseline (*p* = 0.05). Likewise, the placebo group’s ADAS-Cog score dropped by 1.1 points [71,72].

More than thirty real flavonoids, including kaempferol, quercetin, and isorhamnetin conjugated with glucose or rhamnose, are found in *Ginkgo biloba*. Epiafzelechin, epicatechin, epifisetinidol, epigallocatechin, and epirobinetinidol are the proanthocyanidins. Delphinidin, cyanidin, ginkgolides (A, B, C, and J), bilobalide, and trace amounts of pelargonidin make up the extract’s primary terpenoid component. Table 3 and Figure 3 provide a summary of the phytoconstituents of *Ginkgo biloba* and their diverse functions in the management of AD [51,67,72,73,74,75,76,77,78].

### 3.4. Holy basil

*Ocimum tenuiflorum* is a fragrant perennial plant in the Lamiaceae family that has been cultivated for at least three millennia. It is used for a variety of purposes, including medical ones [79]. *Ocimum* sp. leaf extract contains a wide range of phytochemicals having pharmacological relevance, including flavonoids, tannins, alkaloids, saponin, glycosides, terpenoids, and phenols [80,81]. Phenylpropanoids (eugenol and methyl-isoeugenol), germacrene, sesquiterpene isoprenoid α-farnesene, 1,2,4-triethenyl, ursolic acid, β-caryophyllene, cyclopentane, carvacrol, cyclopropylidene, and β-elemene are examined in an analysis of the alcoholic extract of *O. tenuiflorum* [80].

By blocking IκBα phosphorylation and degradation via activating IκB kinase, ursolic acid has been demonstrated to restrict NF-κB activation activities [82,83]. Furthermore, *O. tenuiflorum* extract significantly reduced tau protein and Aβ in the hippocampus, raised BDNF expressions and GABA levels, and decreased glutamate and acetyl cholinesterase activity, according to an in vivo study conducted with Wistar albino rats given Aβ. Cognitive impairment was lessened, and the architecture of the hippocampus was restored [84,85,86].

Clove basil (*Ocimum gratissimum*) and basil (*Ocimum basilicum*) contain the phenylpropanoid eugenol, which has a wide range of pharmacoactivity with regard to AD and neurodegeneration. It increases the effectiveness of niacin (nicotinic acid; vitamin B3) in treating AD [87,88] and lowers the incidence of dementia and the onset of AD brought on by niacin insufficiency [86].

In Charles Foster albino rats administered scopolamine, eugenol increased acetylcholine levels, decreased acetylcholinesterase activity, and increased total muscarinic receptor [89,90]. In another trial, eugenol therapy resulted in a significant reduction in NO and 8-OHdG levels, inhibition of caspase-3 activity, a drop in Bax protein levels, and an increase in Bcl-2, SOD, and GPx levels after aluminum-induced neurotoxicity in Wistar rats [91]. Additionally, it was demonstrated that the volatile sesquiterpene molecule germacrene inhibits BACE1; specifically, germacrene B interacts with the Val93 residue of BACE1 [92], as predicted by the in silico research. However, it was shown that acetylcholinesterase was strongly inhibited by germacrene D [93,94,95,96]. Table 4 and Figure 4 provide an effectiveness summary of the chemicals isolated from *Ocimum tenuiflorum*.

**Table 4 molecules-30-00922-t004:** *Holy basil*’s phytoconstituents and their respective properties for treating AD.

Bioactive Compound	Structure No.	Investigation Method	Mechanism of Action
Ursolic acid [51,82,83]	**25**, in Figure 3	In vitro (cells (Jurkat, HEK-293, KBM-5, H1299, U937))	Inhibits IκB kinase, which prevents IκBα from being phosphorylated and degraded, therefore suppressing NF-κB.
Eugenol [86,87,88,89,90,91]	**26**, in Figure 3	In vitro (cells (SH-SY5Y))	Eugenol enhances the benefits of niacin in the treatment of AD by helping to reduce AB42 levels and the deposition, accumulation, and plaque formation of the protein.
		In vivo (Swiss albino mice; human APP Overexpressing Drosophila transgenic model)	
		In silico	The acetylcholine level increased while the acetylcholinesterase activity decreased (interacting via H-bond with Tyr124, pi-pi T-shaped with Tyr337, pi-pi staked with Ty341, and pi-alkyl interactions with His447, Trp286, Phe295, and Val294).
		In vivo (Charles Foster albino rats)	Reduces acetylcholinesterase activity, increases total muscarinic receptors and acetylcholine level, mitigates glutamate neurotoxicity (increase in glutamate, calcium, calcium-dependent calpain-2, and BDNF), and reduces mitochondrial dysfunction in rats given scopolamine.
		In vivo (Wistar rats)	Decreases the levels of NO and 8-OHdG, blocks the activity of caspase-3, decreases the levels of Bax protein, and raises the levels of Bcl-2, SOD, and GPx in rats that were exposed to aluminum-induced neurotoxicity.
Niacin [86,87,88]	**27**, in Figure 3	In vitro (cells (SH-SY5Y))	By helping to reduce AB42, its deposition, accumulation, and plaque development, eugenol enhances the benefits of niacin in the treatment of AD.
		In vivo (Swiss albino mice; Drosophila transgenic model)	
Germacrene B [51,92]	**28**, in Figure 3	In silico	Inhibits BACE1 by interacting with the BACE1 residue Val93.
Germacrene D [93,94,95,96]	**29**, in Figure 3	In silico	Inhibits acetylcholinesterase.

### 3.5. Zingiberaceae (Ginger Family)

The Zingiberaceae family includes the perennial rhizomatous plant genus *Curcuma*, also known as turmeric. Numerous plant species in this genus are used in ethnomedicine for their therapeutic qualities [97]. The majority of the medicinal actions of turmeric are attributed to its active components, which are called curcuminoids and comprise curcumin and bisdemethoxycurcumin [98]. Several species of *Curcuma*, including *C. zedoaria*, *C. angustifolia*, and *C. leucorrhea*, contain the beneficial phytochemical curcumin. According to reports, *C. longa* has the highest concentration of curcumin [97,98].

Curcumin reduces the production and extension of Aβ fibrils in a dose-dependent manner and destabilizes preexisting Aβ fibrils. Molecular insight into this suppression is provided by an experiment that showed that β-Secretase 1 (BACE1) mRNA levels may be completely reduced at dosages as low as 20 μM [51,99]. Additionally, in another 6-month randomized, double-blind, placebo-controlled clinical study with 34 patients, curcumin was found to reduce and/or reverse the aggregation of Aβ at 0.2 to 1 μM (IC50) [100].

Table 5 lists natural substances that are derivatives of curcumin and have acetylcholinesterase inhibitory properties, such as curcuzedoalide, 3-hydroxy-6-methylacetophenone, and α-curcumene. These compounds are relevant for the therapeutic management of AD. Additionally, the chemicals zedoaraldehyde, 13-hydroxygermacrone, and germacrone were identified from *C. xanthorrhiza*; *Zingiber officinale* contains α-sesquiphellandrene and ar-curcumene; and zerumin A, a diterpene found in labdane, has been isolated from *C. amada* and has been shown to be a strong GABA_A_ receptor modulator [101,102]. Table 5 and Figure 4 provide a summary of the phytoconstituents of Zingiberaceae and their unique roles in the management of AD.

**Table 5 molecules-30-00922-t005:** Zingiberaceae’s phytoconstituents and their respective properties for treating AD.

Bioactive Compound	Structure No.	Investigation Method	Mechanism of Action
α-zingiberene rhizome and leaves [103,104,105]	**30**, in Figure 4	In vitro (by TLC)	Active acetylcholinesterase inhibitor.
ar-curcumene [106,107,108]	**31**, in Figure 4	In vitro (by TLC)	Active acetylcholinesterase inhibitor.
Curcumin [99,100]	**32**, in Figure 4	In vitro	Inhibits Aβ40 and Aβ42 production, extension, and destabilization.
		In vitro (cells (PC-12))	BACE1 mRNA levels mediated by metal ions are totally suppressed at 20 μM.
		In vitro (THP-1 cells)	Restricts the activation of Egr-1 and inhibits the Aβ40-induced production of TNF-α, IL-1β, MCP-1, MIP-1b, and IL-8.
		In vivo (human trial)	At 0.2 to 1 μM (IC50), inhibits and/or reverses the aggregation of Aβ.
Bisdemethoxy-curcumin [109,110,111]	**33**, in Figure 4	In vitro (PBMC isolated from venous blood of AD patients)	Increases the expression of TLR and MGAT3, and changes the phagocytic pathway that removes Aβ plaque.
Zedoaraldehyde [51,112]	**34**, in Figure 4	In vitro (cell HEK293 lines)	Promotes SIRT1 expression.
		In vitro (by TLC)	Exhibits a modest level of acetylcholinesterase inhibition.
13-hydroxy-germacrone [112,113]	**35**, in Figure 4	In vitro (cell HEK293 lines)	Promotes SIRT1 expression.
		In vitro (by TLC)	Exhibits a modest level of acetylcholinesterase inhibition.
Germacrone [106,114]	**36**, in Figure 4	In vitro (cell HEK293 lines)	Promotes SIRT1 expression.
		In vitro (by TLC)	Exhibits a modest level of acetylcholinesterase inhibition.
α-curcumene [51,115,116]	**37**, in Figure 4	In vitro (by TLC)	Exhibits a modest level of acetylcholinesterase inhibition.
3-hydroxy-6-methylaceto- phenone [112,117]	**38**, in Figure 4	In vitro (cell HEK293 lines)	Promotes SIRT1 expression.
Curcuzedoalide [51,118]	**39**, in Figure 4	In vitro (RAW264.7 cells)	Inhibits the synthesis of NO in lipopolysaccharide-stimulated RAW 264.7 cells (IC50: 23.44 ± 0.77 μg/mL)
Zerumin A [112,119]	**40**, in Figure 4	In vitro	Positive modulator of GABA_A_ receptors.

### 3.6. Waterhyssop (Brahmi, Bacopa Monnieri)

The potential advantages of the triterpenoid saponin bacoside-A, which was derived from *Bacopa monnieri* (L.) Wettst, for AD have been the subject of multiple clinical investigations [120,121]. Bacoside A3 may improve cognitive function in AD patients, according to a study. Using an in vitro acetylcholinesterase enzyme inhibitory assay, the bacoside A3’s acetylcholinesterase enzyme inhibitory properties were examined [122]. With IC50 values ranging from 19.4 to 42.8 µg/mL, the extract, fractions, and bacoside-A3 were shown to have a respectable level of acetylcholinesterase inhibitory activity [122].

The safety and efficacy of *Bacopa monnieri* extract on the memory and cognitive capacities of healthy individuals are described in another study. This study, which had 80 healthy participants divided into 40 groups, was double-blind, parallel, and placebo-controlled. The individuals were given 300 mg of the extract, which included 90 mg of total bacosides, every morning after breakfast for 12 weeks, or a placebo. Memory (baseline, days 28, 56, and 84) and cognitive functions (baseline, days 1, 14, 28, 56, and 84) were assessed using the Creyos battery of tests; anxiety and sleep quality were assessed using the Pittsburgh Sleep Quality Index and the Beck Anxiety Inventory at baseline and days 28, 56, and 84. Cortisol levels (baseline, days 56 and 84) and brain-derived neurotrophic factor (day 84) were measured in serum samples. Safety was assessed during the trial using clinical laboratory tests, physical examinations, adverse event tracking, and vital sign monitoring [123,124,125].

According to the study’s findings, 38 participants in the extract group and 36 people in the placebo group completed the trial. The *Bacopa monnieri* extract group showed notable gains in memory (verbal short-term memory, spatial short-term memory, working memory, visuospatial working memory, and episodic memory) and cognitive skills (concentration, alertness, reasoning, and mental flexibility) from baseline to day 84. Cognitive skills were affected as early as day 14, and memory was affected as early as day 28. Additionally, it was demonstrated that ingesting a single dosage of *Bacopa monnieri* extract could significantly impair concentration as soon as three hours later. Comparing the *Bacopa monnieri* extract group to the placebo group on days 28, 56, and 84, the former demonstrated a significant reduction in both anxiety and sleep quality [123,126,127].

Apart from bacoside A3, bacoside A, which includes bacopaside II, bacopaside X, and bacopasaponin C, has been demonstrated to exhibit inhibitory effects on Aβ42-mediated cytotoxicity. Figure 4 and Table 6 provided a summary of bacopasides’ efficacy.

*Bacopa monnieri* may provide additional significant phytochemicals, including stigmasterol and betulinic acid [128]. In Wistar rats, betulinic acid boosts hippocampal long-term potentiation amplification and improves cognition [129]. The ability of betulinic acid to cross the blood–brain barrier and its safety at dose concentrations over 500 mg/kg make it an intriguing therapeutic alternative for long-term use [130,131]. By reducing γ-secretase expression, BACE activity, cholesterol and presenilin distribution in lipid rafts, and BACE1 internalization to endosomal compartments, stigmasterol, a phytosterol, prevented the formation of Aβ plaque [132]. By changing SOD, CAT, glutathione, and GPx, as well as reducing lipid peroxidation in the rat brain, the plant extract has been shown to be successful in changing the oxidative profile of cells [128].

**Table 6 molecules-30-00922-t006:** *Bacopa monnieris*’s phytoconstituents and their respective properties for treating AD.

Bioactive Compound	Structure No.	Investigation Method	Mechanism of Action
Bacoside A3 [122,123]	**41**, in Figure 4	In vitro (cells (SH-SY5Y))	Inhibition of membrane contacts, self-assembly, and membrane disruption to prevent Aβ42-mediated cytotoxicity.
		In vitro (cells (U87MG))	Reduces ROS and prevents NF-κB, iNOS, and PGE2 from moving into the nucleus through the Aβ-mediated pathway (by blocking the overexpression of COX-2).
Bacopaside II Bacopaside X [51,127,133]	**42** and **43** in Figure 4	In vitro (cells (SH-SY5Y))	Inhibition of membrane contacts, self-assembly, and membrane disruption to prevent Aβ42-mediated cytotoxicity.
Bacopasaponin C [127,134]	**44**, in Figure 4	In vitro (cells (SH-SY5Y))	Inhibition of membrane contacts, self-assembly, and membrane disruption to prevent Aβ42-mediated cytotoxicity.
Stigmasterol [128,129,130,131,132]	**45**, in Figure 4	In vitro (C57BL/6J mice)	Decreases BACE activity, BACE1 internalization to endosomal compartments, presenilin and cholesterol distribution in lipid rafts, γ-secretase expression, and Aβ plaque formation.
		In vivo (cells (SH-SY5Y))	
Betulinic acid [129,130,131]	**46**, in Figure 4	In vivo (Wistar rats)	Enhances cognitive function and amplifies long-term potentiation of the hippocampus at a molar ratio of 1:14 (betulinic acid to Aβ).

### 3.7. Tea Plant

*Camellias* belong to the family Theaceae. *Camellia* leaves, which have been eaten for about 500 years, contain a variety of pharmacologically active compounds. Polyphenols, catechins, flavonoids, vitamins, caffeine, amino acids, and polysaccharides are some of these compounds [135]. The phenolic acid theogallin (3–galloylquinic acid) (2–3% dry weight), the amino acid theanine (4–6% dry weight) [136,137], the derivative of carotenoid theaspione [138,139], quamoreokchaside I–II, kamoreokchaside I [140], and phenolic compounds found in black tea, such as theaflavins, thearubigins, and flavonol glycosides [141], are among the other substances that are specific to tea.

Flavones and catechins have been shown to be efficient in reducing Aβ-induced BACE1 mRNA expression in the study conducted on SH-SY5Y neuroblastoma cells. The most successful compounds were epicatechin and epigallocatechin [142]. The flavins’ capacity to scavenge radicals is supported by the antioxidative pharmacophores that their structure produces [143]. It has also been demonstrated that theaflavin-3,3’-gallate and theaflavin-3,3’-digalate block PAI inhibitors, with the latter being the most successful [144]. Both (−)-epicatechin gallate and (−)-epigallocatechin gallate were shown to block the GABA_A_ receptor in an experimental setup in which cRNAs of the α1 and β1 subunits of the bovine GABA_A_ receptors were microinjected into *Xenopus laevis* oocytes [145]. Additionally, black tea extract (polyphenols (442.17 mg/100 g gallic acid equivalent)), theaflavin (2.16%), thearubigins (19.31%), catechins (2.04%), caffeine (1.81%), and theanine (4.1 mg/100 mL) were found to modulate and reduce cellular antioxidant profile like GPx, CAT, and SOD in the hippocampus and cortex of Wistar rats given an aluminum chloride-induced (100 mg/kg, i.p., 60 day) AD model. Additionally, the expression of hallmark proteins Aβ42, AβPP, β-secretase, and γ-secretase was suppressed. [51,146]

L-theanine specifically reduced oxidative damage, inhibited p38 mitogen-activated protein kinase, ERK, and NF-κB, and reduced memory impairment caused by Aβ42 [147]. In a clinical study, 15 elderly patients with MMSE-J scores below 28 were administered 2 g of green tea extract each day, which contained 42 mg of theanine and 227 mg of catechins. The MMSE-J score of the treated group rose by 1.7 points (*p* = 0.03), according to the data [51,148,149,150]. An overview of *Camellia sinensis*’s phytoconstituents and their roles in treating AD is given in Table 7 and Figure 5.

### 3.8. Tinospora cordifolia

Sometimes called Guduchi and heart-leaved moonseed, *Tinospora cordifolia* is a member of the Menispermaceae family [151,152]. The alcoholic extract of *Tinospora cardifolia* was evaluated in relation to step-down-type passive avoidance. It enhanced the plus-maze model for learning and memory in albino mice with amnesia caused by alprazolam at oral dosages of 140 and 280 mg/kg. These extracts made from alcohol made a significant difference. However, 280 mg/kg showed more efficacy than 140 mg/kg and was nearly identical to the standard dosage. According to this study, alcoholic extract from *Tinospora cordifolia* may be advantageous for AD patients [153].

### 3.9. Cissampelos pareira

The effectiveness of the three different oral dosages of *Cissampelos pareira* hydroalcoholic extract—100, 200, and 400 mg/kg—to enhance memory and learning in aged mice with scopolamine-induced amnesia was examined. These extracts decreased acetylcholinesterase activity. A dosage of 400 mg/kg (p.o.) showed a more noticeable benefit in learning- and memory-enhancing activities because of its anti-inflammatory and antioxidant qualities. The results demonstrated *Cissampelos pareira’s* potential significance in the treatment of AD [153,154].

### 3.10. Ficus racemosa

The Moraceae family includes the ornamental plant *Ficus racemosa*. Researchers investigated the memory-enhancing effects of an aqueous extract of *Ficus racemosa* bark at dosages of 250 and 500 mg/kg in rats using the elevated plus-maze model [155,156]. This *Ficus racemosa* extract increased acetylcholine levels in the hippocampus of the experimental mice. The results suggested that *Ficus racemosa* might improve memory [155,156,157].

### 3.11. Moringa oleifera

Using an aqueous extract of *M. oleifera* leaves, the anti-AD action of colchicine (**55** in Figure 5) given orally to rats at a dosage of 250 mg/kg was investigated. This extract demonstrated its effects on the brain by raising levels of catalase and superoxide dismutase (SOD) and lowering lipid peroxidation (LPO) activity in the rat cerebral cortex. The results of the research showed that *M. oleifera* leaves had anti-AD properties [158]. When the homocysteine-induced cognitive impairment and anti-AD activity of the *M. oleifera* extract were examined, it was also demonstrated that the levels of the synaptic proteins PSD95, PSD93, synaptophysin, and synapsin1 were reduced [158,159,160]. The results of the experiment showed that the *M. oleifera* extract exacerbated neurodegeneration and decreased tau hyperphosphorylation [158,159,160].

Three oral dosages of *M. oleifera* hydroalcoholic leaf extract (100, 200, and 400 mg/kg) were tested in a different investigation to see how well they caused neurodegeneration and cognitive impairment in animal models of age-related dementia. The results of the experiment showed that the *M. oleifera* extract greatly increased CAT and SOD activities while dramatically lowering levels of acetylcholinesterase and malondialdehyde (MDA). Other characteristics assessed included memory, neuron density, acetylcholinesterase, and SOD [158,159,160].

### 3.12. Glycyrrhiza glabra

*Glycyrrhiza glabra*, commonly called licorice. The learning and memory effects of an aqueous extract of *Glycyrrhiza glabra* at four different doses—75, 150, 225, and 300 mg/kg—were evaluated in rats with an amnesic paradigm caused by diazepam over the course of six weeks using an oral administration approach. According to the findings, all aqueous extracts of *Glycyrrhiza glabra* enhanced learning and memory tasks [51]. Another study tested the effects of aqueous licorice extract (400 mg/kg) and glabridin (**56** in Figure 5) rich extract (5 and 10 mg/kg) on memory and learning activities against mice’s amnesia caused by scopolamine and diazepam using the oral route of administration. Improvements in memory and learning activities were shown by the results [161,162].

### 3.13. Clitoria ternatea

“Shankpuspi” is the standard term for *Clitoria ternatea*. To investigate the effectiveness of the ethanolic root extract of *Clitoria ternatea* against stress-induced amnesia, rats were administered oral doses of 150 and 300 mg/kg [163,164]. Significant reductions in nitric oxide and DPPH were also seen in this study, in addition to the preventive advantages of *Clitoria ternatea* [164]. In another study, the alcoholic extract of the aerial part and roots of *Clitoria ternatea* was administered to rats at doses of 300 and 500 mg/kg to examine its impact on central cholinergic activity and memory. This extract improved memory performance in the rat brain by increasing acetylcholine levels and acetylcholinesterase activity. Root extract from *Clitoria ternatea* has proven to be more effective than that of aerial components [165].

### 3.14. Sesbania grandiflora

Benzene, acetone, petroleum ether, ethanol, and chloroform extracts of *S. grandiflora* fruits were investigated for cognitive enhancement in rats with dementia caused by a high-fat diet [166]. Significant increases in inhibition were seen for transfer latency, cholesterol, and acetylcholinesterase. To check for any anti-AD effects, mice of various ages were administered 50 and 100 mg/kg of the ethanolic leaf extract and the aqueous flower extract in a distinct experiment. In mice, these extracts restored amnesia, dramatically reduced cholesterol, and inhibited acetylcholinesterase activity [41]. This study evaluated the neuroprotective and neuroprotective effects of *S. grandiflora* seed extract against mice that developed amnesia as a result of celecoxib. MDA and acetylcholinesterase activities were decreased in amnesia-stricken mice, but GSH, SOD, and catalase activities were reversed [167].

### 3.15. Lepidium meyenii

*Lepidium meyenii* is frequently called black maca. At two different oral dosages of 0.5 and 2.0 g/kg, the aqueous extract of *Lepidium meyenii* was investigated for its ability to alleviate memory impairment caused by ovariectomized mice [168]. At different concentrations, chemicals such monoamine oxidase (MAO), acetylcholinesterase, and malondialdehyde (MDA) were measured. According to this study, MDA levels remained constant, whereas acetylcholinesterase and MAO levels were reduced. *Lepidium meyenii* may lessen memory issues, according to the studies [41,168,169].

### 3.16. Nardostachys jatamansi

Using an oral method, the effects of 50, 100, and 200 mg/kg of *Nardostachys jatamansi*’s ethanolic extract on memory and learning were contrasted in young and old mice with amnesia brought on by diazepam and scopolamine. The 200 mg/kg dose dramatically enhanced learning and memory in both young and old mice, as well as rectified the amnesia brought on by scopolamine and diazepam. The potential advantages of *Nardostachys jatamansi* for AD patients were shown in this study [170,171].

In another study, treatments with 200 and 400 mg/kg of methanolic extract *of Nardostachys jatamansi* were utilized to cure cognitive and memory deficits in rats brought on by sleep deprivation. This trial showed a significant improvement in memory and cognitive function in behavioral tests. The methanolic extract of *Nardostachys jatamansi* was shown to have neuroprotective qualities [172,173].

The ethanolic extract of *Nardostachys jatamansi* was examined for amyloid-beta toxicity in both in vitro and in vivo experiments using a Drosophila AD model. The extract from this plant blocked the brain’s Aβ42-induced cell death, reduced the number of glial cells, increased the amount of reactive oxygen species (ROS), and stopped amyloid-β from causing cell death in SH-SY5Y cells. According to the results, *Nardostachys jatamansi* might be a useful herb for treating AD [173,174].

### 3.17. Resveratrol

Red grapes and other plants contain a polyphenol called resveratrol (trans-3,5,4-trihydroxystilbene; **57**, Figure 5), which has been shown to have anti-inflammatory and chemopreventative effects [175,176]. Through the regulation of cytokines and enzymes involved in inflammatory pathways, studies have demonstrated its therapeutic effects in conditions such as inflammatory bowel disease. Resveratrol has strong anti-inflammatory properties through the TLR4/NF-kB/signal transducer and STAT cascade [176,177].

In a double-blind, randomized phase II trial, 119 patients were given either a placebo or resveratrol up to 1000 mg twice daily for 52 weeks in order to treat mild-to-moderate dementia caused by ADs. Despite high dosages, the study discovered minimal levels of resveratrol in cerebral fluid, suggesting limited absorption. The two groups experienced comparable adverse effects, with the resveratrol group reporting higher weight loss. Compared to the placebo group, the resveratrol group’s CSF and plasma Aβ40 levels stabilized, and those with a biomarker-supported diagnosis of AD also experienced a treatment impact on their CSF Aβ42 levels. Although sirtuin engagement was not directly demonstrated by the study, resveratrol may have anti-inflammatory, antioxidant, and anti-Aβ aggregation properties. APOE4 carriers experienced a greater drop in brain volume, while the resveratrol-treated group’s ventricular capacity increased and brain sizes decreased more, according to volumetric MRI. According to these findings, people with mild-to-moderate AD can safely and well tolerate high doses of resveratrol, which may also help to stabilize Aβ levels and halt the loss of brain volume. The mechanisms of action behind resveratrol’s possible benefits in the treatment of AD require more investigation [178,179,180].

Additionally, resveratrol may prevent mouse macrophages and microglial BV-2 cells from reacting to LPS and a TLR4 ligand, according to another study. Additionally, it decreased TNF-α, NO, and IL-6 levels in RAW264.7 cells exposed to LPS. The mRNA and protein expression levels of TLR4 and high-mobility group box 1 (HMGB1) were significantly reduced when cells were treated with LPS plus resveratrol as opposed to LPS alone. Resveratrol (5–20 µM) reduced TNF-α and IL-1, inhibited NF-B activation, and repressed TLR4 expression in cardiomyocytes exposed to anoxia/reoxygenation damage. Treatment improved cell survival after reoxygenation and decreased the immune system’s response to anoxia and reoxygenation damage [181,182,183].

### 3.18. Naringenin

Naringenin (**58**, Figure 5) is a naturally occurring flavonoid molecule found in citrus fruits, especially grapefruit. Naringenin may also have an effect on macrophages. Studies have shown that naringenin can reduce Aβ deposits, cross the blood–brain barrier, and enhance memory function in transgenic AD mice [184]. When primary cultured microglia were treated with Aβ1-42, naringenin administration led to a notable increase in M2 microglia polarization and a decrease in Aβ1-42-induced M1 microglia activation. The brain’s Aβ clearance depends on microglia, which use Aβ-degrading enzymes during phagocytosis. These Aβ-degrading enzymes were up-regulated in M1 microglia and down-regulated in M2 microglia after naringenin therapy. Naringenin thereby increased the Aβ-degrading enzymes in M2 microglia, which might have decreased the production of Aβ plaque [185].

### 3.19. Scutellaria baicalensis

A common medicinal treatment for a variety of acute infectious diseases, such as endotoxins and inflammatory and pyretic conditions, is *Scutellaria baicalensis*, a member of the Lamiaceae family. Its baicalein (**59** in Figure 5) possesses antineuroinflammation characteristics and can down-regulate the toll-like receptor (TLR4) protein in an inflammatory microglial BV2 cell culture [186]. The exposure of BV2 cells to lipopolysaccharides (LPS) results in increased secretion of pro-inflammatory proteins, including tumor necrosis factor-α (TNF-α), interleukin-1β (IL-1β), and interleukin-6 (IL-6), suggesting a significant activation of the cells [187]. The inhibition of the inflammatory response by baicalein is evident. The cytoplasmic NF-κB heterodimer P65-P50 is released into the nucleus upon stimulation of BV2 cells in their resting state. There, it binds to downstream target genes to start target gene transcription [188]. The results of nuclear translocation of NF-κB P65 by immunofluorescence detection showed that the expression of NF-κB p65 protein in the cytoplasm of BV2 cells significantly decreased after LPS stimulation, with the bulk of the protein migrating to the nucleus [188].

Concurrent treatment of baicalein led to a significant down-regulation of the expression of the COX-2 and i-NOS proteins, indicating that baicalein inhibited the inflammation caused by LPS in BV-2 cells. Treatment with baicalein decreased the pro-inflammatory cytokine rise, dopaminergic neuron loss, and motor impairment brought on by MPTP [189]. Baicalein also inhibited the activation of caspase-1 and NLRP3 and decreased gasdermin D (GSDMD)-dependent pyroptosis. Additionally, baicalein inhibited the activation and proliferation of pro-inflammatory microglia associated with disease [189].

### 3.20. Ginseng

As it prevents aging and enhances health, ginseng has been used medicinally for a very long time. Ginsenoside (**60** in Figure 5) Rg1 therapy for AD is linked to improvements in Aβ and tau pathologies, control over synaptic function and intestinal microbiota, and a reduction in inflammation, oxidative stress, and apoptosis [190]. Studies have shown that NOX2 promotes oxidative-stress-induced neuronal damage and aging-related loss of brain function. Additionally, NOX2 expression is significantly elevated in long-cultured hippocampus neurons. Ginsenoside Rg1, the main active component of ginseng, inhibits the activation of the NOD-like receptor protein 1 (NLRP1) and NADPH oxidase 2 (NOX2) inflammasome in hippocampal neurons in vitro, thereby reducing H2O2-induced neuronal damage [191]. ASC generated from inflammation may attach to Aβ in the extracellular space of cells and facilitate Aβ aggregation, which in turn sets off the production of downstream inflammatory components and inflammatory responses, according to research on AD patients and AD transgenic mice. Some studies have suggested that ginsenoside Rg1 has anti-inflammatory qualities [192,193]. Ginsenoside Rg1 can enhance Aβ phagocytosis and inhibit the production of downstream inflammatory factors in the neuroinflammatory response of AD.

### 3.21. Oleuropein

Oleuropein (OLE, **61** in Figure 5), a non-toxic penoid glycoside compound, is mostly found in olive leaves. For three months, researchers administered olive leaf extract continuously to a 5xFAD mouse model. According to studies, oleuropein mainly lowers neuroinflammation by inhibiting the RAGE/HMGB1 pathway, the NF-κB pathway, and the activation of the NLRP3 (NOD-like receptor family with three pyridine domains) inflammasome [194]. Additionally, pretreating SH-SY5Y cells with oleuropein for a whole day may reduce the cell death brought on by copper-Aβ42 and Aβ42. In the transgenic mouse (APPswe/PS1dE9) model, treated animals (OLE) showed a significant decrease in the amount of amyloid plaque development in the cortex and hippocampus when compared to control mice [195].

The phytoconstituents and their individual properties for treating AD of *Tinospora cordifolia*, *Cissampelos pareira*, *Ficus racemosa*, *Moringa oleifera*, *Glycyrrhiza glabra*, *Clitoria ternatea*, *Sesbania grandiflora*, *Lepidium meyenii*, *Nardostachys jatamansi*, resveratrol, naringenin, *Scutellaria baicalensis*, ginseng, oleuropein, and other plants are summarized and outlined in Table 8 and Figure 5.

## 4. The Effects of Combining Natural Remedies for AD

Because they can work in concert to address several disease systems, natural treatments for AD offer a viable therapeutic approach, according to the studies of natural products covered in this research.

Numerous natural remedies, including green tea (EGCG) [141,148], turmeric (curcumin) [99,100], and rosemary (rich in carnosic acid and rosmarinic acid) [58,76], have strong antioxidant qualities, lowering reactive oxygen species (ROS) and increasing the activity of endogenous antioxidant enzymes like SOD, CAT, and GPx [28,136]. The antioxidative effects of these medicines can be enhanced by combining them. For instance, EGCG scavenges ROS, minimizing oxidative-stress-induced neuronal damage, whereas curcumin may disintegrate amyloid-beta plaques. By focusing on mitochondrial redox state, increasing neuronal energy generation, and lowering oxidative burden, rosemary’s rosmarinic acid may work in tandem with curcumin [58,99,136,148].

Eugenol (found in *Holy basil*) [86,87,88,89,90], ursolic acid (found in *Holy basil*) [82,83], and β-caryophyllene (found in rosemary) are examples of compounds that have strong anti-inflammatory effects by modulating microglial activation [58,76], down-regulating NF-κB pathways, and lowering the production of pro-inflammatory cytokines (such as TNF-α and IL-1β). Chronic neuroinflammation can be more successfully suppressed by combining these anti-inflammatory substances. For example, β-caryophyllene stimulates the cannabinoid receptor 2 (CB2), which in turn inhibits glial activation in a synergistic manner with eugenol, which decreases pro-inflammatory signaling [82,90,108].

On the other hand, bacosides from *Bacopa monnieri* and withanolides from ashwagandha encourage autophagy and amyloid-beta clearance [47,48,120], while curcumin prevents amyloid-beta aggregation and destabilizes plaques [99,100]. Therefore, curcumin and bacosides together may be able to remove amyloid plaques more successfully by lowering plaque production and boosting autophagy, the brain’s self-clearing process. Additionally, AChE is inhibited by nepitrin (found in rosemary), bacosides (found in *Bacopa monnieri*), and ginkgolides (found in *Ginkgo biloba*), which raise acetylcholine availability and enhance memory and cognitive performance [48,120]. Combining bacosides with nepitrin can improve cholinergic transmission and have neuroprotective benefits, which may lessen cognitive impairments linked to AD [61,62].

In contrast, bacosides improve synaptic plasticity and encourage the expression of brain-derived neurotrophic factor (BDNF), while withanolides (ashwagandha) stimulate dendritic and axonal development [47,48,120]. Ashwagandha and *Bacopa monnieri* together may promote neuroprotection and cognitive resilience by promoting neuronal repair and connection [48,140]. Additionally, curcumin decreases tau hyperphosphorylation and amyloid-beta aggregation by its multiple action on the tau and amyloid pathways, whereas rosemary components such as carnosic acid limit amyloid elongation and stimulate antioxidant responses. When combined, these substances can simultaneously prevent amyloid aggregation, lessen the creation of tau tangles, and combat oxidative stress [58,77,78].

## 5. Future Directions and Challenges in the Use of Natural Products for AD

A number of intriguing research avenues become apparent as we consider the use of natural ingredients in the treatment of AD. The investigation of particular natural substances with neuroprotective qualities, such as alkaloids and flavonoids, is one important field. These substances have demonstrated the ability to modify additional neurodegenerative processes in addition to blocking acetylcholinesterase. As an illustration of the therapeutic potential found in this field, eugenol, germacrene D, Zingiberaceae, and bacoside-A3 have previously been established as successful natural remedies [88,94,122,201]. Future research should concentrate on how developments in formulation technology are transforming the use of natural products to treat AD, with an emphasis on resolving issues like stability, targeted distribution, and low bioavailability. Natural substances can now be encapsulated to enhance their solubility and controlled release profiles thanks to emerging technologies like nanotechnology-based systems, such as nanoparticles, liposomes, and polymeric micelles [202,203,204]. For example, to improve their BBB penetration and therapeutic efficiency, ginsenosides, resveratrol, and curcumin—all of which are recognized for their neuroprotective and anti-inflammatory qualities—are being repackaged into nanoscale delivery systems [28,180,192]. Furthermore, site-specific distribution is being made easier by sophisticated drug delivery platforms including hydrogels, dendrimers, and solid lipid nanoparticles, which maximize local effects on neuronal tissues while reducing systemic side effects. The accuracy of treatment is being further improved by advancements in biocompatible materials and stimuli-responsive delivery systems that are activated by pH, temperature, or enzymes [202,203,204].

Furthermore, utilizing natural materials for individualized therapy shows promise in customizing therapies for AD treatment depending on genetic and environmental characteristics. Genomic and epigenetic developments are shedding light on how genetic differences, including mutations in the APOE, PSEN1, and PSEN2 genes, affect a person’s susceptibility to AD and how they react to natural substances. In order to determine which patients are most likely to benefit from particular natural items with neuroprotective, anti-inflammatory, or antioxidant qualities, personalized techniques incorporate these genetic findings. Flavonoids, for instance, have demonstrated varying levels of effectiveness based on metabolic pathway changes and genetic predispositions. This suggests that natural product therapies can be maximized by aligning them with contextual variables such as nutrition, lifestyle, and exposure to pollutants, which can also modify AD risk [205,206].

Because natural therapies can target many interrelated pathways in AD, they work in concert. By targeting oxidative stress, inflammation, amyloid-beta accumulation, tau hyperphosphorylation, and synaptic loss all at once, combining these natural substances can increase treatment efficacy. In order to optimize these synergistic benefits while maintaining safety and bioavailability, future research should concentrate on determining the best combinations, doses, and formulations [4,207,208].

Together with developments in pharmacogenomics and personalized medicine, nanoparticles, liposomes, and polymeric micelles hold the potential to customize natural-product-based treatments to each patient’s unique genetic profile, opening the door to safer, more effective, and patient-specific approaches to the fight against AD.

### 5.1. Challenges in Conducting Long-Term, Large-Scale Studies

There are a number of difficulties in carrying out extensive, long-term research to evaluate the therapeutic effectiveness of herbal remedies for AD, especially with regard to patient recruitment, treatment regimen adherence, and the use of suitable outcome measures [209,210].

#### 5.1.1. Patient Recruitment

It is particularly difficult to find a sufficiently large and representative sample of volunteers for AD studies:Perception and awareness: Potential participants and their caregivers might not be aware of clinical trials or may be skeptical of herbal remedies, because they believe they are not as scientifically supported as conventional therapies [210,211].Eligibility rules: The pool of eligible participants may be reduced by strict inclusion and exclusion rules. For example, comorbid conditions may prevent many AD patients from participating [210,211].Study companion requirement: In order to gather data on a participant’s cognitive and functional abilities, AD trials frequently call for a study companion. Finding a trustworthy research partner might be challenging, particularly for patients who do not have regular caregivers [211].

#### 5.1.2. Adherence to Treatment Regimens

It is crucial but difficult to sustain participant adherence throughout long study periods:Cognitive impairment: Participants may find it difficult to adhere to intricate treatment plans due to the cognitive impairments associated with AD, requiring extra assistance from caregivers [209,212].Perceived efficacy and side effects: If participants feel that the herbal medication is not working or if they experience negative side effects, which may worsen over time, they may stop using it [212,213].Complexity of herbal regimens: Herbal therapies can require complicated preparation techniques or several daily dosages, which can make them more difficult for participants and caregivers to follow and perhaps result in lower adherence [210,213].

#### 5.1.3. Outcome Measures

In order to assess the effectiveness of herbal medicines, it is crucial to choose suitable and sensitive outcome measures:Standardization: It can be difficult to standardize treatments and consistently evaluate results, as herbal items can differ in composition, potency, and purity [213,214].Sensitivity to change: More sensitive instruments must be developed or validated, because traditional cognitive tests might not be sensitive enough to identify minor changes brought about by herbal therapies [214,215].Biomarker validation: Although it necessitates substantial research, identifying and validating biomarkers unique to the effects of herbal medicines can help measure treatment efficacy objectively [213,215].

## 6. Conclusions

This study shows that a number of natural medicines derived from plants have considerable promise for treating AD. It has been demonstrated that compounds from *Bacopa monnieri*, such as bacoside A3 and stigmasterol, inhibit amyloid-beta (Aβ) cytotoxicity and reduce plaque formation, while *Camellia sinensis* is rich in polyphenols that effectively reduce Aβ aggregation. These phytochemicals, along with those from *Moringa oleifera*, *Ginkgo biloba*, and *Bacopa monnieri*, have important neuroprotective effects and improve cognitive function. Other important botanicals, such as *Rosmarinus officinalis* and *Tinospora cordifolia*, exhibit benefits that include increasing neurotransmitter levels and improving learning and memory in animal models. The antioxidant and acetylcholine-boosting qualities of *Cissampelos pareira* and *Ficus racemosa* extracts also aid in improving cognitive function.

Additionally, substances such as oleuropein and ginsenoside Rg1 have demonstrated effectiveness in lowering amyloid plaque formation and neuroinflammation, bolstering their potential as therapeutics. These natural compounds’ diverse modes of action, which include acetylcholinesterase inhibition, antioxidant activity, and neuroinflammatory pathway modification, highlight their potential as AD adjunctive therapies.

Overall, the data point to the possibility that by addressing several disease processes, these natural substances and their extracts could provide a comprehensive strategy for treating AD. However, in order to confirm their effectiveness and safety and eventually open the door for their incorporation into all-encompassing AD treatment plans, more investigation and clinical trials are necessary. Furthermore, given the study limitations in this field, we recommend accelerating future studies to enhance the safety and effectiveness of herbal therapy for AD. As a result, treatment approaches will be safer and more successful.

## Figures and Tables

**Figure 1 molecules-30-00922-f001:**
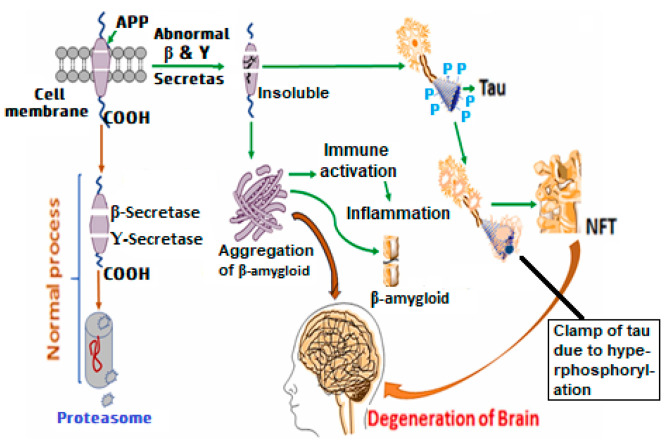
Pathophysiology of AD [15]. Reproduced with permission from Mohd Sajad, Rajesh Kumar, and Sonu Chand Thakur, “History in Perspective: The prime pathological players and role of phytochemicals in AD”; published by ScienceDirect, 2022, licensed by “CC BY 4.0” (https://creativecommons.org/licenses/by/4.0/, accessed on 9 February 2025).

**Figure 2 molecules-30-00922-f002:**
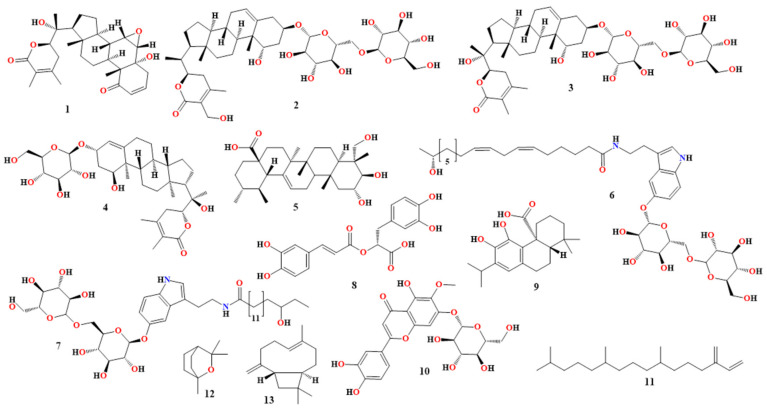
Phytoconstituent chemical structures of ashwagandha and rosemary as mentioned in Table 1 and Table 2.

**Figure 3 molecules-30-00922-f003:**
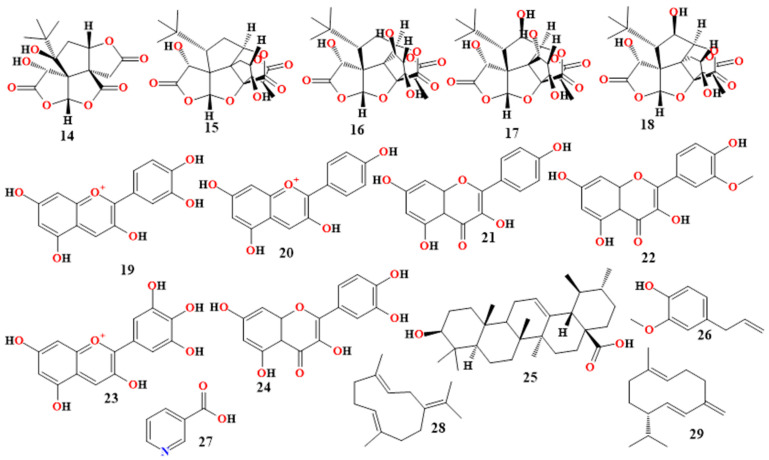
Phytoconstituent chemical structures of *Ginkgo biloba* and *Holy basil* as mentioned in Table 3 and Table 4.

**Figure 4 molecules-30-00922-f004:**
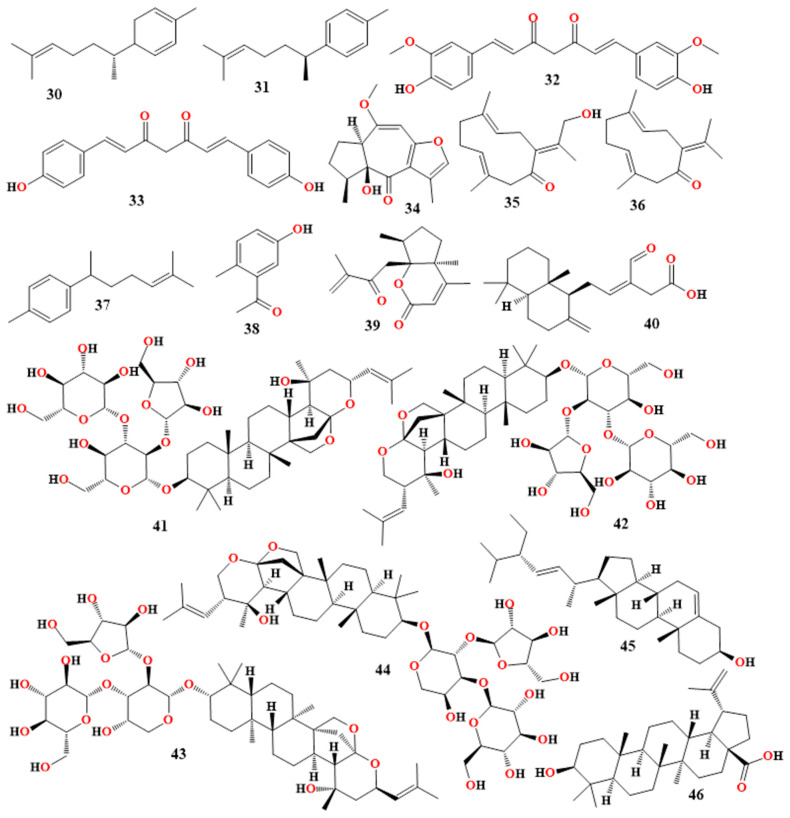
Phytoconstituent chemical structures of Zingiberaceae and *Bacopa monnieris* as mentioned in Table 5 and Table 6.

**Figure 5 molecules-30-00922-f005:**
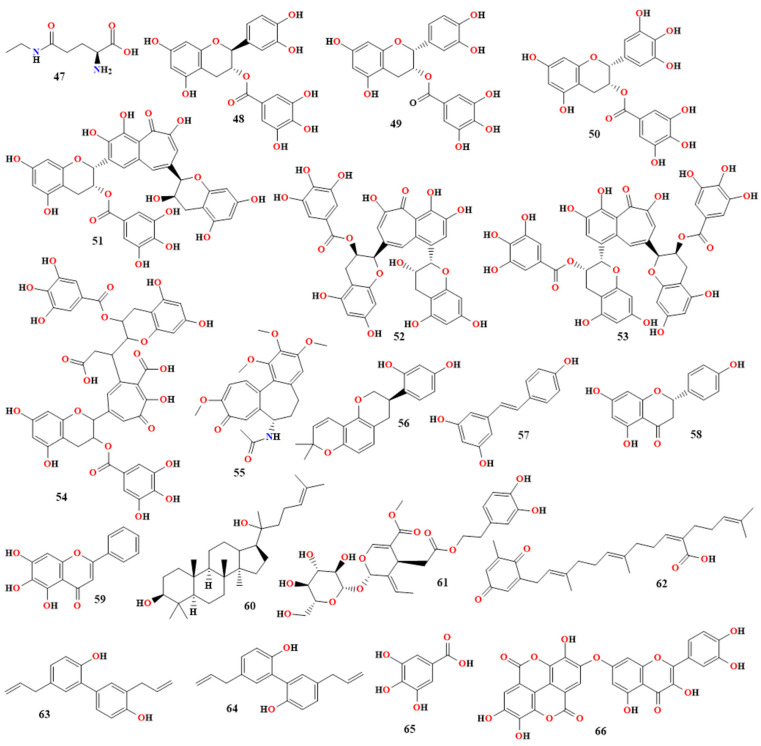
Phytoconstituent chemical structures of *Camellia sinensis*, *Tinospora cordifolia*, *Cissampelos pareira*, *Ficus racemosa*, *Moringa oleifera*, *Glycyrrhiza glabra*, *Clitoria ternatea*, *Sesbania grandiflora*, *Lepidium meyenii*, *Nardostachys jatamansi*, resveratrol, naringenin, *Scutellaria baicalensis*, ginseng, and oleuropein.

**Table 1 molecules-30-00922-t001:** Ashwagandha’s phytoconstituents and their respective properties for treating AD.

Bioactive Compound	Structure No.	Investigation Method	Mechanism of Action
Withanolide A Root extract [49,50,51,52]	**1**, in Figure 2	In silico	Inhibits acetylcholinesterase (intermolecular forces with amino acid residues of the catalytic site).
		In vitro (cell line (SH-SY5Y))	Axonal growth that is not dependent on nerve growth factors (when combined with withanoside IV and VI).
		In vitro (Sprague Dawley rat’s primary cortex neurons)	Increases ADAM10 and decreases BACE1.
Withanoside IV Root extract [49,51,53]	**2**, in Figure 2	In vitro (cell line (SH-SY5Y))	Dendrite growth that is not dependent on nerve growth factors (when combined with withanoside VI and withanolide A).
		In vitro (Sprague Dawley rat’s primary cortex neurons)	Becomes sominone during metabolism, which is thought to encourage axonal development in neurons challenged by Aβ(25–35).
		In vitro (Sprague Dawley rat’s primary cortex neurons)	By phosphorylating the RET (GDNF receptor), sominone promotes the development of new neurons.
		In vitro (cell line (SH-SY5Y))	Increases neurite development (at a dose of 1 mM).
Withanoside VI Root extract [49,51,53]	**3**, in Figure 2	In vitro (cell line (SH-SY5Y))	Dendrite growth that is not dependent on nerve growth factors (when combined with withanoside IV and withanolide A).
		In vitro (cell line (SH-SY5Y))	At a dosage of 1 mM, stimulates neurite development.
Coagulin Q Root extract [51,54]	**4**, in Figure 2	In vitro (cell line (SH-SY5Y))	At a dosage of 1 mM, stimulates neurite development.
Asiatic acid [51,55]	**5**, in Figure 2	In vitro (Sprague Dawley rat’s primary cortex neurons)	Increases ADAM10 and decreases BACE1.
Withanamide A Fruit extract [51,56]	**6**, in Figure 2	In vitro (cells (PC-12))	Provides 100% cell survival at doses of 100 and 50 μg/mL against Aβ-induced cell damage.
		In silico	Bound between the amino acid residues Met35 and Gly25 of the Aβ(39–42) peptide’s active motif.
Withanamide C Fruit extract [51,56]	**7**, in Figure 2	In vitro (cells (PC-12))	Provides 100% cell survival at doses of 100 and 50 μg/mL against Aβ-induced cell damage.
		In silico	Bound between the amino acid residues Met35 and Gly25 of the Aβ(39–42) peptide’s active motif.

**Table 3 molecules-30-00922-t003:** *Ginkgo biloba*’s phytoconstituents and their respective properties for treating AD.

Bioactive Compound	Structure No.	Investigation Method	Mechanism of Action
Bilobalide [51,68]	**14**, in Figure 3	In vivo (Wistar rats)	5HTIA receptors are decreased.
		In vivo (human trial)	Improves GERRI and ADAS-Cog scores.
		In vivo (human trial)	Improvements in senile primary degenerative DAT and MID are seen, as well as a decrease in theta activity in the EEG.
		In vivo (cells (PC12))	Encourages the production of the cytochrome oxidase component COX III and shields mitochondria from the uncoupling of OXPHOS.
		In vitro (rat cerebellar neurons)	Protective effect based on dosage against neuronal death caused by glutamate.
Egb-761: Ginkgolide A Ginkgolide B Ginkgolide C Ginkgolide J Cyanidin Pelargonidin Kaempferol Isorhamnetin [67,68,69].	**15**–**22**, in Figure 3, respectively	In vitro (Wistar rats)	5HTIA receptors are decreased.
		In vivo (human trial)	Improves GERRI and ADAS-Cog scores
		In vivo (human trial)	Improvements in senile primary degenerative DAT and MID are seen, as well as a decrease in theta activity in the EEG.
		In vitro (rat cerebellar neurons)	Protective effect based on dosage against neuronal death caused by glutamate.
		In vivo (gerbil)	Decreases levels of COX III mRNA, which prevents ischemia-induced neuronal death.
		In vitro (cells (PC12))	The ND1 subunit of NADH dehydrogenase mRNA is increased twofold.
Delphinidin [51,63]	**23**, in Figure 3	In vitro (Wistar rats)	5HTIA receptors are decreased.
		In vivo (human trial)	Improvements in senile primary degenerative DAT and MID are seen, as well as a decrease in theta activity in the EEG.
		In vitro (cells (PC12))	GSK-3β inhibition prevents tau hyperphosphorylation.
Quercetin [51,63]	**24**, in Figure 3	In vitro (rat cerebellar neurons)	Protective effect based on dosage against neuronal death caused by glutamate.
		In vivo (human trial)	Improves GERRI and ADAS-Cog scores
		In vivo (gerbil)	Decreases levels of COX III mRNA, which prevents ischemia-induced neuronal death.
		In vitro (cells (PC12))	The ND1 subunit of NADH dehydrogenase mRNA is increased twofold.

**Table 7 molecules-30-00922-t007:** *Camellia sinensis*’s phytoconstituents and their respective properties for treating AD.

Bioactive Compound	Structure No.	Investigation Method	Mechanism of Action
Theanine [147]	**47**, in Figure 5	In vivo (albino Wistar rats)	Suppresses apoptosis (inhibits cytochrome c release, expression of Bax, caspases-3, 8, and 9, promotes Bcl-2 expression), oxidative stress (increasing GPx, CAT, SOD), acetylcholinesterase, BACE1, and γ-secretase, and decreases Aβ load.
		In vivo (Slc:ICR mice)	Decreased oxidative damage, p38 mitogen-activated protein kinase, ERK, and NF-κB, and attenuated Aβ42-induced memory impairment.
		In vivo (human trial)	MMSE-J score improved (dosage: 2 g/day for 3 months; contains 227 mg of catechins and 42 mg of theanine).
(-)-catechin gallate [51,150]	**48**, in Figure 5	In vitro (cell (SH-SY5Y))	Reduces Aβ aggregation at a 10 μg/mL concentration.
(-)-epicatechin gallate [145,146]	**49**, in Figure 5	In vitro (cell (SH-SY5Y))	Reduces Aβ aggregation at a 10 μg/mL concentration.
		In vitro (oocytes from *Xenopus laevis* microinjected with cRNAs encoding the α1 and β1 subunits of the GABA_A_ receptors in cows)	GABA_A_ receptor inhibition (IC50 = 5.5 µM).
(-)-epigallocatechin gallate [51,145,146]	**50**, in Figure 5	In vitro (cell (SH-SY5Y))	Reduces Aβ aggregation at a 10 μg/mL concentration.
		In vitro (oocytes from *Xenopus laevis* microinjected with cRNAs encoding the α1 and β1 subunits of the GABA_A_ receptors in cows)	GABA_A_ receptor inhibition (IC50 = 5.5 µM).
		In silico	Reduces the production of Keap1, which interacts with the protein’s Gly 343, Thr 595, Leu 578, and Asp 579 residues, and encourages Nrf2’s translocation into the nucleus to mitigate the effects of fluoride-induced free radicals.
Theaflavin-3-gallate Theaflavin-3′-gallate Theaflavin 3,3′-digallate [148,149]	**51**–**53** Figure 5	In vitro (erythrocyte ghost prepared from Wistar rat blood)	Protection of erythrocytes against oxidative stress in vitro by molecules’ pharmacophores, which support the molecule’s antioxidative characteristics.
Thearubigin [149]	**54**, in Figure 5	In vitro (albino Wistar rats)	Suppresses apoptosis (inhibits cytochrome c release, expression of Bax, caspases-3, 8, and 9, promotes Bcl-2 expression), oxidative stress (increasing GPx, CAT, SOD), acetylcholinesterase, BACE1, and γ-secretase, and decreases Aβ load.

**Table 8 molecules-30-00922-t008:** *Tinospora cordifolia*, *Cissampelos pareira*, *Ficus racemosa*, *Moringa oleifera*, *Glycyrrhiza glabra*, *Clitoria ternatea*, *Sesbania grandiflora*, *Lepidium meyenii*, *Nardostachys jatamansi*, resveratrol, naringenin, *Scutellaria baicalensis*, ginseng, and oleuropein phytoconstituents and their respective properties for treating AD.

Family Name	Plant Name	Phytoconstituents	Activities	Study Model	Refs.
Menispermaceae	*Tinospora* *Cordifolia*	Flavonoids	Memory enhancement	Albino mice	[151]
Menispermaceae	*Cissampelos* *Pareira*	Alkaloids	Antiamnesic activity	Mice	[154]
Moraceae	*Ficus racemosa*	Tannins, saponins	Anticholinesterase activity	Wistar rats	[155,156,157]
Moringaceae	*Moringa oleifera*	Proteins, fatty acid	Antioxidant, anticholinesterase activity	Wistar rats	[158,159,160]
Fabaceae	*Glycyrrhiza glabra*	Glabridin, volatile oils	Antiamnesic	Mice	[162]
Fabaceae	*Clitoria ternatea*	Tannins, glycosides, flavonoids	Neuroprotective, antioxidant, anticholinergic activity	Rats	[163,164,165]
Fabaceae	*Sesbania grandiflora*	Tannins, gums	Neuroprotective, antidiabetic, antioxidant, antidementia activity	Mice	[166,167]
Brassicaceae	*Lepidium* *Meyenii*	Proteins	Anticholinesterase and antioxidant activity	Mice	[168]
Caprifoliaceae	*Nardostachys* *Jatamansi*	Sesquiterpenes	Antioxidant, inhibition of Aβ accumulation	Mice	[170,171,172,173]
Vitaceae	*Vitis vinifera*, grape	Resveratrol	Decreases Aβ oligomer toxicity; inhibits neuronal autophagy; reduces apoptosis; prevents inflammation; stops tau from being phosphorylated	Mice	[175,183]
Araliaceae	*Panax ginseng*	Ginsedosides	Memory enhancement	Human	[190,191,192,193]
Phyllanthaceae	*Emblica officinalis*	Gallic acid, ellagic acidquercetin (**65**, **66** in Figure 5)	Antiamnesic, anticholinesterase, and antioxidant	Mice	[196]
Celastraceae	*Celastrus paniculatus*	Alkaloids, sesquiterpenes	Acetylcholinesterase, BchE inhibitory activity and antioxidant activity	Mice	[197]
Moraceae	*Ficus carica*	Flavonoids, phenolic compounds, vitamins	Antioxidant and immunostimulant activity	Wistar rats	[198]
Myristicaceae	*Myristica fragrans*	Terpenes, flavonoids	Antioxidant, memory enhancement, acetylcholinesterase inhibitor	Mice	[199]
Sargassaceae	*Saragassum sagamianum*	Sargaquinoic acid (**62** in Figure 5) plastoquinones and sargamchromenols	Acetylcholinesterase and BchE inhibitor	Human	[200]
Mangnoliaceae	*Magnolia officinalis*	Honokiol, magnolol (**63**, **64** in Figure 5)	Antioxidant, anticholinesterase, neuroprotective	Transgenic mice	[200]

## Data Availability

This article has compiled the plant-derived chemicals that have an effect on Alzheimer’s disease along with their sources in Table 1, Table 2, Table 3, Table 4, Table 5, Table 6 and Table 7.

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
