# Peer review of "Promising Natural Remedies for Alzheimer’s Disease Therapy"

_molecules, 2025, doi:10.3390/molecules30040922_

Round 1
Reviewer 1 Report
Comments and Suggestions for Authors
The manuscript by Thawabteh et al. provides a comprehensive review of the potential of natural remedies in the treatment of Alzheimer's disease (AD). The authors have successfully compiled a vast array of herbal extracts and natural compounds that exhibit efficacy in enhancing memory and cognitive performance, which is a significant contribution to the field. The review is timely and aligns with the growing interest in alternative therapies for AD. The manuscript is generally well-structured and well-written, but there are areas that could benefit from further refinement.
1. The introduction effectively sets the stage for the review by outlining the prevalence and impact of AD, as well as the limitations of current treatment strategies. It would be beneficial to include a brief discussion on the criteria used to select the natural remedies discussed in the review.
2. The section on the pathophysiological features of AD is well-detailed. It would be advantageous to include a comparative analysis of the mechanisms of action between the natural products discussed and conventional AD treatments.
3. The discussion on various phytochemicals is informative. The authors might consider adding a section on the potential synergistic effects of combining different natural remedies, as this could be a promising area for future research.
4. While the manuscript highlights the potential benefits of natural remedies, there is a need for a more in-depth discussion on their safety profiles, including any reported side effects or toxicities associated with long-term use.
5. The authors suggest the need for larger-scale studies with longer treatment durations. It would be valuable to discuss potential challenges in conducting such trials, including patient recruitment, adherence to treatment regimens, and outcome measures.
6. The conclusion effectively summarizes the main points of the review. However, it could be strengthened by providing specific recommendations for future research directions based on the gaps identified in the current literature.
7. Others:there are “:” or “.” after the subtitles 3.1-3.21. Please format it.
Author Response
First of all, we would like to thank the reviewer for his fruitful comments
Comments and Suggestions for Authors
The manuscript by Thawabteh et al. provides a comprehensive review of the potential of natural remedies in the treatment of Alzheimer's disease (AD). The authors have successfully compiled a vast array of herbal extracts and natural compounds that exhibit efficacy in enhancing memory and cognitive performance, which is a significant contribution to the field. The review is timely and aligns with the growing interest in alternative therapies for AD. The manuscript is generally well-structured and well-written, but there are areas that could benefit from further refinement.
- The introduction effectively sets the stage for the review by outlining the prevalence and impact of AD, as well as the limitations of current treatment strategies. It would be beneficial to include a brief discussion on the criteria used to select the natural remedies discussed in the review.
Response: The comments were addressed in lines 99- 113 and are highlighted with a green.
- The section on the pathophysiological features of AD is well-detailed. It would be advantageous to include a comparative analysis of the mechanisms of action between the natural products discussed and conventional AD treatments.
Response: The necessary information can be found in the third section's tables, where each plant under study has its mode of action (separate column) listed next to it. Kindly review tables 1–7. The publisher (MDPI) may view the rewriting of each plant's or compound's mechanisms and comparative analysis as a rehash of the same material that doesn't add anything new.
- The discussion on various phytochemicals is informative. The authors might consider adding a section on the potential synergistic effects of combining different natural remedies, as this could be a promising area for future research.
Response:
- We have added a new section (4) “The Effects of Combining Natural Remedies for AD”
- Section 4 became Section 5: Future direction and challenges of the natural product for AD
- The added text in sections 4 and 5 are marked in blue.
- While the manuscript highlights the potential benefits of natural remedies, there is a need for a more in-depth discussion on their safety profiles, including any reported side effects or toxicities associated with long-term use.
Response: While we acknowledge the value of this material, the goal of the publication is to "provide a thorough overview of the most promising natural products for the treatment of AD."
- The authors suggest the need for larger-scale studies with longer treatment durations. It would be valuable to discuss potential challenges in conducting such trials, including patient recruitment, adherence to treatment regimens, and outcome measures.
Response: We have added a new sub-section (5.1) with is highlighted in red.
- The conclusion effectively summarizes the main points of the review. However, it could be strengthened by providing specific recommendations for future research directions based on the gaps identified in the current literature.
Response: We have added the reviewer’s recommendations and they are marked with orange.
- Others:there are “:” or “.” after the subtitles 3.1-3.21. Please format it.
Response:
Summary:
Figure 1. Pathophysiology of AD, the font size has been increased.
Line 99-113: green font, the criteria used to select the natural remedies discussed in the review.
Line 739-779: blue font, the Effects of Combining Natural Remedies for AD
Line 779: Section 4 became Section 5: Future direction and challenges of the natural product for AD
Line 812-817: blue font, future vision for the synergy between natural remedies arises from their ability to target multiple interconnected pathways in AD.
Line 822-863: red font, challenges in conducting long-term, large-scale studies
Line 885-887: orange font, specific recommendations for future research directions
Line 1410-1433: New references (228-235) have been added to the manuscript, after adding some paragraphs requested by the reviewers, highlighted with yellow
Reviewer 2 Report
Comments and Suggestions for Authors
The manuscript is a valuable review of ntural products which show therapeutic potential in AD treatment. The article is well written. The figures could be somehow modify to to make them more clear by alignment of the structures and numbers.
Author Response
The manuscript is a valuable review of natural products which show therapeutic potential in AD treatment. The article is well written. The figures could be somehow modify to make them more clear by alignment of the structures and numbers.
Response:
We thank the reviewer for his beneficial comments.
Yes, excellent comment. As the reviewer suggested, the publisher will be advised to reformat before publishing. The publishing office is responsible for preparing the final figures.
Reviewer 3 Report
Comments and Suggestions for Authors
The article “Promising natural remedies for Alzheimer's disease therapy” by Mahmood Thawabteh et al. presents an interesting literature from 2015-2024 on natural substances and their effects on AD. The article is well-written and the data are clearly presented and discussed in a manner that is both comprehensive and accessible. The article can be published after addressing some of the minor issues as givn below:
1. The text in Figure 1. is not visible. Increase the font size. Introduce the bad amyloids specific number of amino aicds, such as 1-42 or 1-40.
2. In introduction section, also add a small section of vaccine treatment, antibody treatment options for the AD that run under clinical trials OR already failed in clinical trials.
3. Authors should add a section related to the diagnostic testing available so far for the occurance of AD by checking the pathological biomarkers.
4. It would be helpful for the readers to understand the mechanism along with the structures, if the structures given in Figures are incorporated in their respective tables. It will also help to reduce the length of the article.
The article is already polished, after addressing these minor issues, the article is good for publishing in the journal Molecules.
Author Response
First of all, we would like to thank the reviewer for his fruitful comments
Comments and Suggestions for Authors
The article “Promising natural remedies for Alzheimer's disease therapy” by Mahmood Thawabteh et al. presents an interesting literature from 2015-2024 on natural substances and their effects on AD. The article is well-written and the data are clearly presented and discussed in a manner that is both comprehensive and accessible. The article can be published after addressing some of the minor issues as givn below:
- The text in Figure 1. is not visible. Increase the font size. Introduce the bad amyloids specific number of amino aicds, such as 1-42 or 1-40.
Response: Done
- In introduction section, also add a small section of vaccine treatment, antibody treatment options for the AD that run under clinical trials OR already failed in clinical trials.
- Authors should add a section related to the diagnostic testing available so far for the occurrence of AD by checking the pathological biomarkers.
Response 2&3: This document succeeded in achieving its objective, which was to "provide a thorough overview of the most promising natural products for the treatment of AD." Furthermore, by examining pathological biomarkers and vaccine and antibody treatment options for AD that are undergoing clinical trials or have already failed in clinical trials, we covered in detail the diagnostic testing currently available for the occurrence of AD in our two earlier manuscripts that were published in the same journal. The US government database, which we analyzed, found clinical trials of vaccines, small molecule medications, and biologics for 14 agents in Phase I, 34 in Phase II, and 11 in Phase III that may be finished by 2028.It would be helpful for the readers to understand the mechanism along with the structures, if the structures given in Figures are incorporated in their respective tables. It will also help to reduce the length of the article.
We appreciate your comment, but not all of the compounds in the figures are listed in the tables. Some, like colchicine, glabridin, resveratrol, naringenin, and others, are mentioned in the text while it is being written. Additionally, when put in tables, large compounds like 2, 3, 7, 10, 41–44, and 51–54 may become less clear. We therefore prefer to be in figures rather than tables for these reasons.
Others:
Response:
Line 99-113: green font, the criteria used to select the natural remedies discussed in the review.
Line 739-779: blue font, the Effects of Combining Natural Remedies for AD
Line 779: Section 4 became Section 5: Future direction and challenges of the natural product for AD
Line 812-817: blue font, future vision for the synergy between natural remedies arises from their ability to target multiple interconnected pathways in AD.
Line 822-863: red font, challenges in conducting long-term, large-scale studies
Line 885-887: orange font, specific recommendations for future research directions
Line 1410-1433: New references (228-235) have been added to the manuscript, after adding some paragraphs requested by the reviewers, highlighted with yellow
Round 2
Reviewer 1 Report
Comments and Suggestions for Authors
The responses answered the questions. The paper is good to go from my view.